# Infinite Ergodic Walks in Finite Connected Undirected Graphs [note 1]

**DOI:** 10.3390/e23020205

**Published:** 2021-02-08

**Authors:** Dimitri Volchenkov

**Affiliations:** Department of Mathematics and Statistics, Texas Tech University, 1108 Memorial Circle, Lubbock, TX 79409, USA; dimitri.volchenkov@ttu.edu

**Keywords:** statistical ensembles of walks, entropic force and pressure, graph’s navigation, graph node’s navigability, graph node’s fugacity, 02.10.Ox, 05.40.Fb, 05.20.Gg

## Abstract

The micro-canonical, canonical, and grand canonical ensembles of walks defined in finite connected undirected graphs are considered in the thermodynamic limit of *infinite walk length*. As infinitely long paths are extremely sensitive to structural irregularities and defects, their properties are used to describe the degree of structural imbalance, anisotropy, and navigability in finite graphs. For the first time, we introduce entropic force and pressure describing the effect of graph defects on mobility patterns associated with the very long walks in finite graphs; navigation in graphs and navigability to the nodes by the different types of ergodic walks; as well as node’s fugacity in the course of prospective network expansion or shrinking.

## 1. Introduction

The precursor of a concept of statistical ensembles and the related ergodic hypothesis formulated by Boltzmann [1,2] were met with a violently negative reaction by the great majority of scientists for clumsiness, absurd, and paradoxical consequences [3], although it allowed the theoretical calculation of the equations of state for the first time. The study of statistical ensembles related to graphs and networks suffers from a similar inhospitable reception from scientists playing cup-and-ball with a swarm of heuristic parameters and giving any importance to their connection with each other, which is often responsible for spurious conclusions on the graph’s structure and function. The thermodynamic approach to graphs was initiated in complex network theory concerned with the thermodynamic limit of infinitely large graph size N→∞ [4], in which a graph’s structural “fluctuations” become negligible. The major result of the theory on structurally homogeneous infinite graph (random trees) is the Bose–Einstein condensation mechanism explaining the growth of complex evolving networks as a topological phase transition between a “rich-get-richer” phase and a “winner-takes-all” phase [5,6,7]. In contrast to complex network theory, we consider the statistical ensembles of walks defined on a finite connected undirected graph in the thermodynamic limit of very long walks n→∞, which has previously never been addressed. Statistics of lengthy walks elucidates the graph structure, quantifies navigability of the graph, and evaluates the fugacity of graph nodes with respect to the entire system of infinite paths available in the graph—all of these characteristics are introduced and discussed in our work for the first time. The probability measuring the tendency of a graph to shrivel or expand at a node follows the Fermi–Dirac distribution function. Although we have sketched a set of “ideal gas laws” for the structure of networks and graphs (in the last section of our work), we have not formulated a comprehensive structural "equation of sate" for graphs and networks yet.

The probability we assign to an event depends on whether we count it as *one of many*, considered all at once, or as a single event of its kind. In other words, an estimated likelihood of events hinges on their assumed membership in an *ensemble* described by some probability distribution. The famous *Two-Child Paradox* [8] serves a good example for this point: “*Mr. Smith has two children. At least one of them is a boy. What is the probability that both children are boys?*” Given that a child is either a boy (*B*) or a girl (*G*) with equal probability 12, two incompatible answers may be given to this question, depending on the assumptions taken.

On the one hand, as the probability of getting a boy equals Pr(B)=12 uniformly and unconditionally for all families, using the Bayes’ Theorem, we obtain that the probability of having at least one boy in a two-kid family will be the same as just having a boy, viz.,
PrB&B|B=PrB|B&B×Pr(B&B)Pr(B)=1×1412=12.
On the other hand, as having a boy in an *ensemble* of two-child families with at least one boy obviously comprises three possible events, i.e., B&B, or G&B, or B&G, the probability of getting a boy in a family of two equals Pr(B)=Pr(B&B)+Pr(B&G)+Pr(G&B)=14+14+14=34, and therefore
PrB&B|B=PrB|B&B×Pr(B&B)Pr(B)=1×1434=13.
The ensemble interpretation, in which each admissible event in a family of two with a boy appears *equally probable*, is preferable in the context of *ergodic hypothesis* blind to family history. In the context of the Two-Child Paradox, there is no way in probability theory to discern if the gender composition in such a family stays put, or children change their sex exploring possible gender identities during an infinite lifetime provided at least one of them stays a boy. The ergodic hypothesis helps to avoid this awkward question by equating the ensemble and time averages while replacing a dynamic description of identity changes by the probabilistic description within the ensemble over a very long period of time. Switching temporal and ensemble perspectives under the spell of ergodic hypothesis is assumed in thermodynamics, equilibrium statistical mechanics, and the theory of dynamical systems.

The concepts of ensembles and ergodicity have a long history [3]. Boltzmann introduced a "*monode*", a family of possible *stationary probability distributions* over a single cyclic trajectory of a system of gas particles on an energy surface in the phase space as early as in 1844 [1,2]. According to the Boltzmann hypothesis (Equation 1), the time spent by a system in some region of the phase space is proportional to the volume of this region, so that all accessible microstates are *equiprobable* over a long period of time, viz.,
(1)$$\lim_{T\to \infty }\frac{dt}{T}=\frac{\sigma ds}{\int \sigma ds},$$
where σ is the probability density of microstates on the iso-energetic surface, whose area element is ds. With this hypothesis, Boltzmann [1,2] and later Helmholtz [9,10] were able to explain the classical equilibrium thermodynamics, which successfully describes the behavior of gases. The concept of *thermodynamic ensembles* was further developed and coined into the English-speaking world by Gibbs [11].

In our work, we review three classical thermodynamic ensembles defined by Gibbs [11]—the *microcanonical* (Section 2), *canonical* (Section 4), and *grand canonical* (Section 8) ensembles of very long walks defined in finite connected undirected graphs—and demonstrate that the concept of ergodic ensembles might be applied to quite abstract objects of discrete mathematics. The *thermodynamic limit* in our approach is defined as the limit of *very long* walks n→∞ in a *finite* graph rather than the limit for a large number of graph nodes *N*. In the limit N→∞, “fluctuations” of graph structural features are negligible, and therefore the graph can be considered as structurally homogeneous across all scales—random, in the limit n→∞ fluctuations of the growth rate of the number of distinguishable, long walks in the graph can be ignored, and then graph’s *topological entropy*
$\mu =\log_{2}\alpha_{\max}$ (the log of graph’s *spectral radius*) and the corresponding Perron eigenvector of the graph adjacency matrix describe the degree of structural complexity, anisotropy, and navigability of the graph.

Each thermodynamic ensemble permits specific statistical behavior. For example, the microcanonical ensemble representing an *isolated* system (with constant energy) is defined by assigning equal probability to every walk of a given length existing in the graph. All very long walks that fit some *probability distribution* over graph’s nodes constitute a *macrostate* in the canonical ensemble of walks defined in the graph. For example, the series of *intrinsic random walks* (introduced in Section 5) make up equal probabilities to all walks of a given length starting at a node providing an example of the canonical ensemble of walks defined on the finite graph. This canonical ensemble contains not only the very well-known isotropic nearest-neighbor random walks on finite graphs [7,12], but also infinitely many types of less known *anisotropic* random walks on graphs—and the *Ruelle–Bowen random walk* [13,14] making up all infinite walks starting at each node equally probably is one among them. While the ergodic theory for isotropic random walks on finite graphs is well developed [15,16] (We profoundly thank our referee for this remark), the ergodic properties of anisotropic random walks, including their statistical confinement in the best structurally integrated sub-graphs (see Section 5 and Section 7), have not been discussed in literature yet. Finally, in an *open* system of long walks represented by the grand canonical ensemble, *chemical potential* (free energy absorbed by a very long walk seizing graph’s edge) is kept fixed and equal the graph’s topological entropy μ.

We also discuss applications of ergodic walks to the structural analysis of and navigation through finite undirected connected graphs. Graph’s structural defects and boundaries repel very long walks that can be be expressed in terms of entropic pressure and force (Section 3). Intrinsic random walks forming the canonical ensemble in a graph can be used to measure the degree of graph’s structural anisotropy (Section 5), to estimate the amount of predictable (*navigable*) information about present navigator’s location (Section 6) and assess the *navigability* to each graph node in proportion to its relative *visiting frequency* (Section 7). Navigation focuses on locating a navigator’s position compared to known locations, paths, and structural patterns [17]. The navigability to a node comprises two information components compatible with two major navigation strategies, known as *path integration* (that allows for keeping track of the position and heading while exploring a new space) and *landmark-based piloting* (re-calculating position when in a familiar environment), working in concert during navigation in humans and animals [17]. Finally, the grand canonical ensemble describes the statistics of local fluctuations of the growth rate of the numbers of long walks around the chemical potential as n→∞ (Section 8). The distribution of these fluctuations follows Fermi–Dirac statistics and marks graph’s defects and boundary nodes hosting dramatically less very long walks than others.

We conclude in the last section.

## 2. The Micro-Canonical Ensemble of Equiprobable Walks in Finite Connected Undirected Graphs

The number of walks of length *n* (i.e., *n*–walks) in a lattice $Z^{d}$ in *d*-dimensional space grows exponentially with *n*, $N_{n}=2^{nd}$. The *micro-canonical ensemble* is defined by assigning *equal* probability to every *n*–walk, viz.,
(2)℘n=12nd=exp−nd1ln2≡expFnkT,
where the (Boltzmann constant and) temperature kT≡1ln2, the *free energy* of the *n*–walks is
(3)$$F_{n}\equiv -\log_{2}{℘}_{n}=-kT\;\ln N_{n}\equiv -kT\;H_{n}=nd,$$
and $H_{n}\equiv \ln N_{n}=nd\ln2$ is the *entropy* in a micro-canonical ensemble. As the free energy $F_{n}$ is the Legendre transformation of the *internal energy*
$U_{n}$, with kT as the independent variable [18,19], viz.,
$$F_{n}\equiv U_{n}-kTH_{n},$$
comparing this definition with (Equation 3), we conclude that the internal energy of all *n*–walks is $U_{n}=0$ in a micro-canonical ensemble.

We also readily extend the statistical description of micro-canonical ensemble of equiprobable walks to κ-regular graphs, in which every vertex has the same number of neighbors, $\kappa =2^{d}$, by using the substitution $d=\log_{2}\kappa$. As the free energy value (Equation 3) grows linearly with *n*, the *intensive free energy* (per absorbed edge), viz.,
(4)$$\mu \;\;\equiv \;\;\lim_{n\to \infty }\frac{F_{n}}{n}\;\;\;=\;\;kT\lim_{n\to \infty }\frac{H_{n}}{n}\;=\;\log_{2}\kappa \;=\;d,$$
plays the role of *chemical potential* describing the change to free energy after absorbing a new edge to a very long walk in a κ-regular graph in a micro-canonical ensemble.

Given a finite connected undirected graph G(V,E) where *V*, $\left|V\right|=N$, is a set of vertices, and E⊆V×V is a set of edges, we assume that its *adjacency matrix* (such that $A_{i,j}=1$, $\left(i,j\right)\in E$, and $A_{i,j}=0$, otherwise) has the following spectral decomposition $A_{ij}=\sum_{s=1}^{N}\alpha_{s}u_{is}u_{sj},$ with ordered eigenvalues $\alpha_{\max}\equiv \alpha_{1}>\alpha_{2}\geq \cdots \geq \alpha_{N}$. The free energy in the micro-canonical ensemble equals
(5)$$\begin{array}{ccc} F_{n} & = & \log_{2}N_{n}=\log_{2}\;\sum_{ij}A_{ij}^{n}=\log_{2}\;\sum_{ij}\left(\sum_{s=1}^{N}\alpha_{s}^{n}u_{is}u_{js}\right)\\ \; & = & \log_{2}\sum_{s=1}^{N}\alpha_{s}^{n}\gamma_{s}^{2}=\log_{2}\gamma_{1}^{2}\alpha_{\max}^{n}\left(1+\sum_{s=2}^{N}\frac{\gamma_{s}^{2}}{\gamma_{1}^{2}}{\left(\frac{\alpha_{s}}{\alpha_{\max}}\right)}^{n}\right),\;\gamma_{s}\equiv \sum_{i=1}^{N}u_{is}, \end{array}$$
and, since αsαmax<1, the intensive free energy amounts to the logarithm of the spectral radius $\alpha_{\max}$ of the graph, viz.,
(6)$$\mu =\lim_{n\to \infty }\frac{F_{n}}{n}=\lim_{n\to \infty }\frac{1}{n}\log_{2}\gamma_{1}^{2}\alpha_{\max}^{n}\left(1+\sum_{s=2}^{N}\frac{\gamma_{s}^{2}}{\gamma_{1}^{2}}{\left(\frac{\alpha_{s}}{\alpha_{\max}}\right)}^{n}\right)=\log_{2}\alpha_{\max}\equiv d_{G}.$$

In Section 8, the quantity (Equation 6) plays the role of *chemical potential* of an edge absorbed by a very long walk. For a κ—regular graph, its spectral radius $\alpha_{\max}=\kappa$, so that ${\mu =\log}_{2}\kappa =d,$ in accordance with (Equation 4). The log of graph spectral radius (Equation 6) is also called the *topological entropy* of the graph [20,21] because it is the exponential growth rate of the number of distinguishable walks, being a measure of complexity of the graph structure. According to (Equation 4), the topological entropy of the graph μ can also be interpreted as the *effective dimension of space* of the graph, $d_{G},$ in a micro-canonical ensemble of very long walks.

## 3. Entropic Pressure and Force in Micro-Canonical Ensemble of Walks

Missing nodes and edges might dramatically reduce the number of very long walks available in a graph, reshaping the global mobility patterns in a micro-canonical ensemble of walks. Statistical changes in mobility patterns due to graph defects that can be described in terms of *entropic pressure* and *entropic force* are as follows.

Namely, a missing node depletes the number of very long walks available in the graph, and therefore reduces the corresponding free energy, $F_{n}=\log_{2}\;\sum_{ij}A_{ij}^{n},$ by the following amount of *local energy*,
(7)$$\begin{array}{ccc} E_{i}^{\left(n\right)} & = & \log_{2}\sum_{j}{\left(A^{n}\right)}_{ij}\\ \; & = & \log_{2}\alpha_{\max}^{n}u_{i1}\gamma_{1}\left(1+\sum_{s=2}^{N}{\left(\frac{\alpha_{s}}{\alpha_{\max}}\right)}^{n}\left(\frac{u_{is}}{u_{i1}}\right)\left(\frac{\gamma_{s}}{\gamma_{1}}\right)\right), \end{array}$$
corresponding to the number of very long walks anchoring at *i*, viz.,
(8)$$\delta_{i}F_{n}\equiv \left(F_{n}-E_{i}^{\left(n\right)}\right).$$

In the thermodynamic limit n→∞, the resulting local increment of free energy measuring its sensitivity to the disappearance of node from the graph is as follows:(9)$$\begin{array}{ccc} {\left.\Delta F\right|}_{i} & = & \lim_{n\to \infty }\left(\delta_{i}F_{n}-F_{n}\right)=-\lim_{n\to \infty }\;\log_{2}\left(\frac{\sum_{ij}{\left(A^{n}\right)}_{ij}-\sum_{j}{\left(A^{n}\right)}_{ij}}{\sum_{ij}{\left(A^{n}\right)}_{ij}}\right)\\ \; & = & -\log_{2}\left(1-\frac{u_{i1}}{\gamma_{1}}\right)\equiv P_{i}. \end{array}$$

We call the resulting quantity (Equation 9) *entropic pressure*
$P_{i}$, as it accounts for the local stress characterizing the transfer of walker’s mobility from *i* to the rest of the graph if *i* is not available (see Figure 1 Left).

Similarly, by eliminating an edge $\left(i,j\right)\in E$ from the graph, we reduce the local energy $E_{i}^{\left(n\right)}$ of the node i∈V (Equation 7) by the following amount, $\delta E_{i,j}^{\left(n\right)}=\log_{2}\left(\sum_{s}A_{is}^{n}-A_{ij}\left[\sum_{k}{\left(A^{n-1}\right)}_{jk}\right]\right)\;,$ corresponding to the number of $\left(n-1\right)$-walks available from the node j∈V adjacent to *i*, viz.,
(10)$$\begin{array}{ccc} {\left.\Delta E\right|}_{i\to j} & = & \lim_{n\to \infty }-\log_{2}\left(\frac{\sum_{s}{\left(A^{n}\right)}_{is}-A_{ij}\sum_{k}{\left(A^{n-1}\right)}_{ik}}{\sum_{s}{\left(A^{n}\right)}_{is}}\right)\\ \; & = & \lim_{n\to \infty }-\log_{2}\left(1-\frac{A_{ij}\sum_{k}{\left(A^{n-1}\right)}_{jk}}{\sum_{s}{\left(A^{n}\right)}_{is}}\right)\\ \; & = & -\log_{2}\left(1-\frac{A_{ij}u_{j1}}{\alpha_{\max}u_{i1}}\right)=-\log_{2}\left(1-W_{ij}^{\left(\infty \right)}\right)\equiv F_{ij}. \end{array}$$

The direction dependent *entropic force*
$F_{ij}$ introduced in (Equation 10) emerges from the statistical tendency of very long walks to follow the *preferential transition*
$W_{ij}^{\left(\infty \right)}$ to the neighboring nodes hosting many infinitely long walks, as in the Ruelle–Bowen random walk (Equation 19) [13,14]. It is worth-mentioning that the expression for the entropic force (Equation 10) has the structure of a Laplacian operator $L_{ij}=\left(1-W_{ij}^{\left(\infty \right)}\right)$ related to random walks defined in the graph *G* by the transition matrix $W_{ij}^{\left(\infty \right)}$.

In Figure 1, we have presented a membrane graph with a defect and highlighted its nodes according to the values of entropic pressure (Equation 9) (left) and the elements of Perron eigenvector of the matrix $F_{ij}$ (Equation 10) (right) in the membrane graph.

In Figure 2, we use the graph representation of Lubbock, TX, USA acquired from the *OpenStreetMap* service (The *OpenStreetMap* database is publicly available at https://dataverse.harvard.edu/dataverse/osmnx-street-networks). To construct the spatial graph of the city, we used *Python’s lxml* library to parse the raw data and obtain the spatial graph adjacency matrix. The data set was cleaned further by removing disconnected neighborhoods, such as the Preston Smith International airport that is not a structural part of the city. The resulting connected city graph of Lubbock contains 10,421 nodes representing all spaces of movement, including but not limited to residential, secondary, tertiary roads, trunk links, and highways.

The value of entropic pressure in the spatial graph of Lubbock attains maximum at the contemporary structural focus of the city, far apart from the city historical downtown (Figure 2 Left). The nodes of the city spatial graph on the right-hand side of Figure 2 are highlighted according to elements of the Fiedler eigenvector belonging to the second largest eigenvalue of the entropic force matrix $F_{ij}$ (i.e., the second smallest eigenvalue of the associated Laplacian matrix $L_{ij}$). The Fiedler eigenvector is used in spectral graph partition, as it bisects the graph into only two connected communities based on the sign of the second vector entry. The Fiedler eigenvector indicates the direction of the fastest decrease of the entropic force over the city spatial graph of Lubbock (Figure 2 Right). The entries of the Fielder eigenvector are zero everywhere, except for a narrow band extended from the historical city center (where the magnitudeof entropic force is positive) toward the contemporary structural focus of the city (where the magnitude of entropic force is negative). The structural focus of the city absorbs very long walks while the historical center anchored at the abolished city railway station expels long walks. Although railway construction enhanced the city status of Lubbock in early days, its maintenance has a continuing negative impact on the urban development, since railways barricade streets, dramatically cutting down the number of possible paths people can drive or walk and create isolated neighborhoods [22].

## 4. The Canonical Ensemble of Walks in Finite Connected Undirected Graphs

The *canonical ensemble* represents the possible states of a system in equilibrium that does not evolve over time, even though the underlying system might be in constant motion [11]. The canonical ensemble is a collection of very long walks (*microstates*) of length $n=\sum_{r=1}^{s}n_{r}\gg 1$, where $n_{r}$ counts the number of visits paid by a walker to the *r*-th vertex of a connected undirected graph G, compatible with a π-*macrostate*, a discrete probability density vector ${\left\{\pi_{r}\right\}}_{r=1}^{N}$, $\sum_{r=1}^{N}\pi_{r}=1$, taken over the set of graph vertices, viz., $n_{r}/n{⟶}_{n\to \infty }\pi_{r}$.

The total number of microstates (i.e., long walks) lumped into a single π-macrostate is then given by the following multinomial coefficient:(11)$$M\left(n,s\right)\;\;=\;\;\frac{n!}{n_{1}!\cdots n_{s}!}=\frac{n!}{\left(n\pi_{1}\right)!\cdots \left(n\pi_{s}\right)!}$$

Using Stirling’s approximation, lnn!≈−n+nlnn, we readily obtain that
(12)$$\begin{array}{ccc} \ln M\left(n,s\right) & \approx & \left(n-n\ln n\right)-\left(n_{1}-n_{1}\ln n_{1}\right)\cdots -\left(n_{s}-n_{s}\ln n_{s}\right)\\ \; & = & n\sum_{r=1}^{s}\frac{n_{r}}{n}\cdot \ln\frac{n_{r}}{n}, \end{array}$$
and therefore, as n→∞
(13)$$M\left(n,s\right)\approx \exp\left(n\sum_{r=1}^{s}\frac{n_{r}}{n}\cdot \ln\frac{n_{r}}{n}\right)\approx \exp\left(n\sum_{r=1}^{s}\pi_{r}\cdot \ln\pi_{r}\right)\approx \exp\left(-nH\right),$$
in which
(14)$$H\;\;\equiv \;\;-\sum_{r=1}^{N}\pi_{r}\cdot \ln\pi_{r},\;0\cdot \log0=\log0^{0}=\log1=0,$$
is the *Boltzmann–Gibbs–Shannon entropy* [23,24] in the canonical ensemble. If every very long walk lumped to the π-macrostate is chosen with equal probability among the other walks suited for the same macrostate, ${℘}_{n}\approx \exp nH$, then the most probable walks would be those compatible with the uniform density πr=1N,r=1,…,N, maximizing the value of entropy (Equation 14), $H_{\max}=\ln N.$ The free energy over the canonical ensemble of very long π-walks (n→∞) is given by
(15)$$F_{n}\;\;=\;\;-kT\ln{℘}_{n}\approx kT\cdot nH,$$
and, therefore, the intensive free energy (chemical potential) equals
(16)$$\mu =\lim_{n\to \infty }\frac{F_{n}}{n}=kT\cdot H=-\sum_{r=1}^{N}\pi_{r}\cdot \log_{2}\pi_{r}\equiv I\left(\pi \right),$$
where $I\left(\pi \right)$ is the amount of *information* (in bits) revealed at every step of the π -walk.

## 5. The Canonical Ensemble of Intrinsic Random Walks in Finite Connected Undirected Graphs

Discrete time *random walks*
$W=\left\{X_{n}\in V:n\in Z\right\}$ defined in a finite connected undirected graph G(V,E) by an irreducible row -stochastic transition probability matrix $W_{ij}=$$\mathrm{Pr}\left(X_{n+1}=j|X_{n}=i\right)>0$, i,j∈V,$\;\left(i,j\right)\in E$ are the natural candidates for the π-macrostates in the canonical ensemble of walks. Indeed, as the row-stochastic transition matrix $W_{ij}$ does not evolve over time, the unique stationary distribution of the random walk W is the major left eigenvector ${\pi =\left\{\pi_{r}\right\}}_{r=1}^{N}$ of the transition matrix, such that $\sum_{s=1}^{N}\pi_{r}W_{rs}=\pi_{s}$.

Given the graph adjacency matrix $A_{i,j}=1,$$\left(i,j\right)\in E,$ and $A_{i,j}=0$ otherwise, we define the $n^{th}$-*order degree* of the vertex i∈V as the number of *n*-walks available at i∈V, viz.,
(17)$$\kappa_{i}^{\left(n\right)}\equiv \sum_{j=1}^{N}{\left(A^{n}\right)}_{ij},\;\kappa_{i}^{\left(0\right)}=1.$$

Taking further into account that $\kappa_{i}^{\left(n+1\right)}=\sum_{j=1}^{N}A_{ij}\kappa_{j}^{\left(n\right)}$, we derive an infinite sequence of transition probability matrices [25], viz.,
(18)$$W_{ij}^{\left(n\right)}=\frac{A_{ij}\kappa_{j}^{\left(n\right)}}{\kappa_{i}^{\left(n+1\right)}}=\frac{A_{ij}\sum_{s=1}^{N}{\left(A^{n}\right)}_{js}}{\sum_{s=1}^{N}A_{is}\sum_{s=1}^{N}{\left(A^{n}\right)}_{sr}},\;\;\;\;\sum_{j=1}^{N}W_{ij}^{\left(n\right)}=1,\;n\in N,$$
defining a countable set of *intrinsic random walks* in the graph *G*.

The first order intrinsic random walk defined by the transition matrix Wij1=Aijκi1 has been discussed in literature for more than a century [12,26]. The walk $W_{ij}^{\left(1\right)}$ is locally *isotropic*, as the random walker chooses the next node to visit among all nearest neighbors of the current node with equal probability. In Figure 3, we presented densities of nodes in the membrane graph with respect to the different types of intrinsic random walks. Density of nodes with respect to $W_{ij}^{\left(1\right)}$ is proportionate to their *degree centrality*, i.e., the numbers of links incident upon the nodes (Figure 3, left). Other intrinsic random walks following the transition probabilities, $W_{ij}^{\left(n\right)}$, n>1, make all $\kappa_{i}^{\left(n\right)}$*n*-walks starting at the node *i* to occur with equal probability. These random walks are locally biased (*anisotropic*), as transitions to the nearest neighbors providing more lengthy walks are more preferable under (Equation 18) for n>1 [25]. In the limit n→∞, the series of transition matrices $W_{ij}^{\left(n\right)}$ converges [25] to the *Ruelle–Bowen random walk* [21] (also known as the *maximal entropy random walk* [27]), viz.,
(19)$$W_{ij}^{\left(\infty \right)}=\lim_{n\to \infty }W_{ij}^{\left(n\right)}=\lim_{n\to \infty }\frac{A_{ij}\kappa_{j}^{\left(n\right)}}{\kappa_{i}^{\left(n+1\right)}}=\lim_{n\to \infty }\frac{A_{ij}\alpha_{\max}^{n}u_{j1}\gamma_{1}}{\alpha_{\max}^{n+1}u_{i1}\gamma_{1}}=\frac{A_{ij}u_{j1}}{\alpha_{\max}u_{i1}}.$$

The anisotropic random walk $W_{ij}^{\left(\infty \right)}$ is confined in the central nodes of the membrane graph (Figure 3, right). The stationary distribution for the intrinsic random walks (Equation 18) reads as follows [25]:(20)$$\pi_{i}^{\left(n\right)}=\frac{\kappa_{i}^{\left(n\right)}\kappa_{i}^{\left(n-1\right)}}{\sum_{s=1}^{N}\kappa_{s}^{\left(n\right)}\kappa_{s}^{\left(n-1\right)}}.$$

For the isotropic random walks $W_{ij}^{\left(1\right)}$, the stationary distribution πi1=κi12E, where *E* is the total number of edges in the graph [12], and $\pi_{i}^{\left(\infty \right)}=u_{i1}^{2}$, for the Ruelle–Bowen random walks [27]. The stationary distribution $\pi_{i}^{\left(1\right)}$ reports on the *degree centrality* of the graph nodes (i.e., the number of links incident upon a node), and $\pi_{i}^{\left(\infty \right)}$ is naturally related to the *eigenvector centrality*
$u_{i1}$ of the node *i* in the graph *G* [28].

The time until a random walk approaches the stationary distribution (Figure 3) (i.e., the mixing time) is determined by the spectral gap, the difference between the two largest eigenvalues of the transition matrix. Spectral gaps is maximum (mixing time is minimum) over the canonical ensemble of intrinsic random walks for the anisotropic random walk $W_{ij}^{\left(\infty \right)}$ (Figure 4).

The *relative entropy rate* [29] between two Markov chains defined by their transition matrices,
(21)$$\begin{array}{ccc} \eta (n) & = & \sum_{i=1}^{N}\pi_{i}^{\left(1\right)}\sum_{j=1}^{N}W_{ij}^{\left(1\right)}\log_{2}\frac{W_{ij}^{\left(1\right)}}{W_{ij}^{\left(n\right)}}=\frac{1}{2E}\sum_{i,j}A_{ij}\log_{2}\frac{\kappa_{i}^{\left(n\right)}}{\kappa_{j}^{\left(n-1\right)}\kappa_{i}^{\left(1\right)}}\\ \; & = & \frac{1}{2E}\sum_{i,j}\left(A_{ij}\log_{2}\frac{\kappa_{i}^{\left(n\right)}}{\kappa_{j}^{\left(n-1\right)}}-\delta_{ij}\log_{2}\kappa_{i}^{\left(1\right)}\right)\equiv \frac{1}{2E}\sum_{i,j}\left(A_{ij}\Delta_{ij}^{\left(n\right)}-\delta_{ij}d_{i}\right), \end{array}$$
can be used for measuring information divergence over the canonical ensemble of intrinsic random walks in connected undirected graphs and the degree of *graph directional anisotropy* [25].

In (Equation 21), we have introduced $d_{i}\equiv \log_{2}\kappa_{i}^{\left(1\right)},$ a local counterpart of the space dimension parameter (Equation 4), and its generalization to *n*-walks, the directional *graph space dimension tensor*
(22)$$\Delta_{ij}^{\left(n\right)}\equiv \log_{2}\frac{\kappa_{i}^{\left(n\right)}}{\kappa_{j}^{\left(n-1\right)}}$$
measuring the degree of *directional anisotropy* in transitions of the intrinsic random walks making up all *n*-walks available from the node *i* with equal probability. For n=1, the graph space dimension tensor (Equation 22) reduces to the space dimension, $\Delta_{ij}^{\left(1\right)}=d_{i}$, as $\kappa_{i}^{\left(0\right)}=1$ for all nodes. In the thermodynamic limit n→∞, the graph space dimension tensor reduces to a *direction dependent* counterpart of the effective space dimension of the graph $d_{G}$ (Equation 6) (or the graph topological entropy), viz.,
(23)$$\Delta_{ij}^{\left(\infty \right)}=\log_{2}\frac{\alpha_{\max}u_{i1}}{u_{j1}}.$$

## 6. Navigation through Graphs over Canonical Ensembles of Walks

The problem of effective navigation in graphs and networks can be considered in the framework of canonical ensemble of walks, since the navigator location prediction requires a density of locations that is known. Frequently visited sites are predicted more efficiently than little frequented, especially in the long-run [30].

Given a π -walk $W=\left\{X_{t}\in V:t\in Z\right\}$ defined in a connected undirected graph G(V,E), Bayes’ theorem [29,31] describes the probability of navigator’s present location *X* based on prior knowledge of her previous location *t* steps before, $X_{-t}\overset{t}{\to }X.$ Namely, $X_{-t}$ may be a *t*-*step precursor* of *X* with the following probability: (24)$$\mathrm{Pr}\left(X_{-t}|X\right)=\frac{\mathrm{Pr}\left(X_{-t}\overset{t}{\to }X\right)\pi \left(X_{-t}\right)}{\pi \left(X\right)},$$
where $\mathrm{Pr}\left(X_{-t}\overset{t}{\to }X\right)$ is the probability of walking from $X_{-t}$ to *X* precisely in *t* steps; $\pi \left(X_{-t}\right)$ and $\pi \left(X\right)$ are the densities of locations $X_{-t}$ and *X* with respect to the π-walk, respectively. $\mathrm{Pr}\left(X_{-t}|X\right)$ is a *density of the*
*t*-*step precursors* for the location *X* induced by the density of walks π. If $\mathrm{Pr}\left(X_{-t}|X\right)=\pi \left(X_{-t}\right),$ it follows from (Equation 24) that the location *X* is *unpredictable* (as any other location $X_{-t}$ is a precursor for *X*). The available information about visiting the location *X* at present is therefore scattered over the entire graph in the past and can be assessed by observing all possible *t*-step precursors $X_{-t}$, viz.,
(25)$$P\left(X\right)=\mathrm{Pr}\left(X_{-t}|X\right)\;\log_{2}\frac{\mathrm{Pr}\left(X_{-t}|X\right)}{\pi (X_{-t})}$$

The information divergence [29] (Equation 25) vanishes if and only if the density of *t*-step precursors $\mathrm{Pr}\left(X_{-t}|X\right)$ for the location *X* over the graph *G* is identical to $\pi (X_{-t})$, so that visiting the location $X_{-t}$ in the past is statistically independent of visiting the present location *X*
*t* steps later, and therefore $X_{-t}$ is not a *t*-step precursor of *X* [30]. The amount of information (Equation 25) attains its *maximum* value, viz.,
(26)$$\max P_{t}\left(X\right)=-\log_{2}\pi \left(X\right),$$
whenever the marginal probability $\pi \left(X\right)$ is the major left eigenvector of the *t*-step transition matrix $\mathrm{Pr}\left(X_{-t}\overset{t}{\to }X\right)$, so that $\mathrm{Pr}\left(X_{-t}|X\right)=1,$ for all *t* and $X_{-t}$, i.e., visiting any location in the graph *G* by π-walk with probability 1 is a predictor for visiting any other location *X*
*t* steps later.

According to the Boltzmann equation (Equation 1), for ergodic observables, the time average of the maximal information (Equation 26) over the entire history of π-walks equals the entropy of the π-walk (Equation 16), viz.,
(27)$$\lim_{t\to \infty }\frac{1}{t}\sum_{\tau =0}^{t}\max P_{\tau }\left(X\right)=-\sum_{\left\{X\right\}}\pi \left(X\right)\log_{2}\pi \left(X\right)=kT\cdot H\left(X\right)\equiv I\left(\pi \right).$$

However, the actual amount of predictable (*navigable*) information about present navigator’s location may be quite modest, much less than the amount information revealed at every step of the π-walk ((Equation 16) and (Equation 27)): different graphs have different degrees of *navigability*.

## 7. Navigability of Graphs and Graph Nodes over Canonical Ensembles of Walks

The information function (Equation 27) can be represented as a sum of the *predictable* and *unpredictable* information components [32], viz.,
(28)$$I\left(\pi \right)=P\left(\pi \right)+U\left(\pi \right).$$

The predictable information component $P\left(\pi \right)$ measures the amount of apparent uncertainty about the navigator’s location that can be resolved with some navigation strategy compatible with the π-walks, and $U\left(\pi \right)$ gauges the *amount of true uncertainty* about the navigator’s location that cannot be inferred anyway. In the following, we attribute the predictable information component $P\left(\pi \right)$ to the *navigability of the graph*
*G* by the π-walk.

Assuming that both information components in (Equation 28) have the same form as the information function (Equation 27), viz.,
(29)$$P\left(\pi \right)=-\sum_{r=1}^{N}\pi_{r}\cdot \log_{2}\varphi_{r},\;\mathrm{and}\;U\left(\pi \right)=-\sum_{r=1}^{N}\pi_{r}\cdot \log_{2}\psi_{r},$$
with some partition functions $\varphi_{r}$ and $\psi_{r}$, such that $\pi_{r}=\varphi_{r}\psi_{r}$, we obtain
(30)Iπ=−∑r=1Nπr·log2φrψr,φr=πrψr.

We call the partition function $\varphi_{r}$ the *navigability to the node*
r∈V in the graph *G* by the π-walk. Obviously, the navigability to the node $\varphi_{r}$ is proportional to its relative visiting frequency $\pi_{r}$—as the more frequent the location, the higher its forecast accuracy—and inverse proportional to the partition function $\psi_{r}$ assessing *uncertainty of visiting the node*
*r* by the π-walk.

There are two major navigation strategies—*landmark-based piloting* and *walk integration*—working in concert during wayfinding in humans and animals [17]. First, the next visit location $X_{t+1}$ can be guessed from the present navigator’s position $X_{t}$ in the graph, and the degree of accuracy of such a guess can be assessed by the mutual information between the present and future navigator’s location conditioned on the walk history, $I\left(X_{t};X_{t+1}|X_{t-1},\ldots X_{1}\right)$. This strategy can be naturally associated with landmark-based piloting.

If the π-walk is a random walk defined by a transition matrix $W_{ij}$, the *conditional mutual information* for such a Markov chain depends only upon the immediate past navigator’s location $X_{t-1}$, but not on the entire historical sequence of locations visited by the navigator in the more distant past [32], so that
(31)$$\begin{array}{ccc} I\left(X_{t};X_{t+1}|X_{t-1}\right) & = & H\left(X_{t+1}|X_{t-1}\right)-H\left(X_{t}|X_{t-1}\right)\\ \; & = & \sum_{k=1}^{N}\pi_{k}\sum_{r=1}^{N}\left(W_{kr}\log_{2}W_{kr}-W_{kr}^{2}\log_{2}W_{kr}^{2}\right). \end{array}$$

Second, some degree of uncertainty about the navigator’s future location $X_{t+1}$ might be resolved after all revisiting, and a possible correlation between walks are taken into account in the course of walk integration over the presumably infinite motion history of π-walk. The latter quantity is given by the *excess entropy* [33,34,35],
(32)$$E\left(\pi \right)=I\left(\pi \right)-h\left(\pi \right)$$
where the *entropy rate* [29],
(33)$$h\left(\pi \right)=\lim_{t\to \infty }\frac{1}{t}\sum_{k=1}^{t}H\left(X_{t}|X_{t-1},\ldots X_{1}\right)$$
quantifies the mean amount of uncertainty consisting in the whole (infinite) path history of the π-walks. However, it is intuitive that the values of conditional entropies $H\left(X_{t}|X_{t-1},{\ldots X}_{1}\right)$ in the r.h.s. of (Equation 33) do not increase with the length of walks and, therefore, $I\left(\pi \right)\geq h\left(\pi \right)$, so that $E\left(\pi \right)\geq 0$.

For a random walk defined by a transition matrix $W_{ij}$, the Markov property simplifies the expression for the entropy rate (Equation 33), viz.,
(34)$$\begin{array}{ccc} h\left(\pi \right) & = & \lim_{t\to \infty }\frac{1}{t}\sum_{k=1}^{t}H\left(X_{t}|X_{t-1},\ldots X_{1}\right)\\ \; & = & \lim_{t\to \infty }\left[\frac{1}{t}H\left(X_{1}\right)+\frac{1}{t}\left(H\left(X_{2}|X_{1}\right)+H\left(X_{3}|X_{2}\right)+\ldots \right)\right]\\ \; & = & \lim_{t\to \infty }\left[\frac{1}{t}H\left(X_{1}\right)+\frac{t-1}{t}H\left(X_{2}|X_{1}\right)\right]\\ \; & = & H\left(X_{2}|X_{1}\right)=-\sum_{k=1}^{N}\pi_{k}\sum_{r=1}^{N}W_{kr}\log_{2}W_{kr}, \end{array}$$
so that the excess entropy (Equation 32) reads as follows:(35)$$E\left(\pi \right)\;\;=\;\;-\sum_{k=1}^{N}\pi_{k}\cdot \left(\log_{2}\pi_{k}+\sum_{r=1}^{N}W_{kr}\log_{2}W_{kr}\right).$$

By summing (Equation 31) and (Equation 35), we obtain the total amount of predictable information P(π) revealed by the π-walk in the graph *G*, viz.,
(36)$$\begin{array}{ccc} P(\pi ) & = & E\left(\pi \right)+I\left(X_{t};X_{t+1}|X_{t-1}\right)\\ \; & = & -\sum_{k=1}^{N}\pi_{k}\cdot \left(\log_{2}\pi_{k}+\sum_{r=1}^{N}W_{kr}^{2}\log_{2}W_{kr}^{2}\right)\\ \; & = & -\sum_{k=1}^{N}\pi_{k}\cdot \log_{2}\left(\pi_{k}\cdot \prod_{r=1}^{N}W_{kr}^{2W_{kr}^{2}}\right), \end{array}$$
so that *the navigability to the node*
*r* in the graph *G* by the random walks defined by the transition matrix $W_{ij}$ is
(37)$$\varphi_{k}=\pi_{k}\cdot \prod_{r=1}^{N}W_{kr}^{2W_{kr}^{2}}\;$$

Navigability to a node evaluated by the partition function (Equation 37) depends on the strategy of walkers. In Figure 5, we illustrate the difference by highlighting the nodes of the membrane graph according to the degrees of navigability by the isotropic random walks $W_{ij}^{\left(1\right)}$ (left) and anisotropic random walks $W_{ij}^{\left(\infty \right)}$ (right). For the isotropic random walks, the movement of walkers along the low-dimensional boundaries and at the corners of the graph are more predictable than their movements in the bulk, as all bulky locations of the same connectivity are visited with equal probability by the random walk $W_{ij}^{\left(1\right)}$. In contrast with the isotropic random walks, a navigator following the anisotropic strategy $W_{ij}^{\left(\infty \right)}$ is statistically confined within the region hosting the most of infinitely long paths available in the graph, where the navigator’s position is very likely.

As demonstrated in [32], the entropy function $I\left(\pi \right)\equiv H\left(X_{t}\right)$ allows for the following decomposition involving the conditional entropies:(38)$$\begin{array}{cc} H\left(X_{t}\right) & \equiv H\left(X_{t}\right)-H\left(\left.X_{t+1}\right|X_{t}\right)+H\left(\left.X_{t+1}\right|X_{t}\right)\\ \; & =\left(H\left(X_{t}\right)-H\left(X_{t+1}|X_{t}\right)\right)+H\left(X_{t+1}|X_{t}\right)+\left\{H\left(X_{t}|X_{t-1}\right)-H\left(X_{t}|X_{t-1}\right)\right\}\\ \; & \;+\left\{H\left(X_{t+1}|X_{t-1}\right)-H\left(X_{t+1}|X_{t-1}\right)\right\}\\ \; & =\underset{E(\pi )}{\underset{︸}{\left(H\left(X_{t}\right)-H\left(X_{t+1}|X_{t}\right)\right)}}+\underset{I\left(\left.X_{t};X_{t+1}\right|X_{t-1}\right)}{\underset{︸}{\left(H\left(\left.X_{t+1}\right|X_{t-1}\right)-H\left(\left.X_{t}\right|X_{t-1}\right)\right)}}\\ \; & \;+\underset{U\left(\pi \right)}{\underset{︸}{\left(H\left(\left.X_{t+1}\right|X_{t}\right)+\left.H\left(\left.X_{t}\right|X_{t-1}\right)-H\left(\left.X_{t+1}\right|X_{t-1}\right)\right.\right)}}. \end{array}$$

Therefore, the remaining part of the information function (Equation 28), the last part in the decomposition (Equation 38), is the conditional entropy of the present navigator’s location conditioned on her past and future locations, viz.,
(39)$$U\left(\pi \right)=I\left(\pi \right)-P\left(\pi \right)=H\left(X_{t}|X_{t-1}\right)+H\left(X_{t}|X_{t-1}\right)-H\left(X_{t+1}|X_{t-1}\right)$$
assesses the amount of *true uncertainty* about a navigator’s location that can neither be inferred from integrating over the past history of the π-walk nor have any repercussion for the navigator walk in the future [34]. For a random walk defined by the transition matrix $W_{ij}$, we readily obtain that
(40)$$\begin{array}{ccc} U\left(\pi \right)=-\sum_{k=1}^{N}\pi_{k}\cdot \log_{2}\psi_{k} & = & -\sum_{k=1}^{N}\pi_{k}\cdot \log_{2}\left(\frac{\pi_{k}}{\varphi_{k}}\right)\\ \; & = & -\sum_{k=1}^{N}\pi_{k}\cdot \log_{2}\left(\prod_{r=1}^{N}W_{kr}^{-2W_{kr}^{2}}\right), \end{array}$$
where the partition function $\psi_{k}$ assesses the amount of uncertainty about navigator’s visiting the node k∈V.

## 8. A Grand-Canonical Ensemble of Ergodic Walks in Finite Connected Undirected Graphs

The grand canonical ensemble represents the possible states of a system exchanging energy and particles with a heat bath in thermodynamic equilibrium [11]. The growth rate of the number of distinguishable paths in a graph tends to its topological entropy $\mu =\log_{2}\alpha_{\max}$ in the thermodynamic limit n→∞. However, the local growth rate of the number of distinguishable paths available from a node, $\log_{2}\alpha_{\max}\gamma_{1}u_{i1},$ might differ from the graph topological entropy. In the grand canonical ensemble, the probability to observe such a "fluctuation" of the long paths growth rate inferior to the topological entropy at the node *i* is taken to be
(41)$$P_{i}^{\left(n\right)}=\frac{1}{Z_{n}}\exp\left(\frac{\delta_{i}F_{n}}{kT}\right),\;Z_{n}\equiv \sum_{j=1}^{N}\exp\left(\frac{\delta_{j}F_{n}}{kT}\right),$$
where $\delta_{i}F_{n}=\left(F_{n}-E_{i}^{\left(n\right)}\right)$ is a fluctuation of free energy (Equation 8) associated with heterogeneity of growth rate of the number of very long walks in the graph. The *grand partition function*
$Z_{n}$ amasses the *fugacity*
$\exp\left(\frac{\delta_{j}F_{n}}{kT}\right)$ of all nodes in the graph, playing the role of a normalization factor in (Equation 41). In the thermodynamic limit n→∞, $\lim_{n\to \infty }F_{n}=$$\lim_{n\to \infty }\log_{2}\sum_{ij}A_{ij}^{n}$= $\lim_{n\to \infty }\log_{2}\alpha_{\max}^{n}$=nμ, and $\lim_{n\to \infty }E_{i}^{\left(n\right)}$$=\lim_{n\to \infty }\log_{2}\alpha_{\max}^{n}u_{i1}\gamma_{1}$, so that the expression for the *node’s fugacity* takes the following form:(42)expδiFkT=limn→∞expnμ−Ei(n)kT=exp[log2αmaxn−log2αmaxnui1γ1]1ln2=αmaxnαmaxnui1γ1=1ui1γ1,γ1≡∑j=1Nuj1,
the grand partition function reads as follows:(43)$$Z\equiv \lim_{n\to \infty }Z_{n}=\frac{1}{\gamma_{1}}\sum_{j=1}^{N}\frac{1}{u_{j1}},\;\gamma_{1}\equiv \sum_{i=1}^{N}u_{i1},$$
and, finally, the grand canonical probability (Equation 41) takes the form of a Fermi–Dirac distribution in the thermodynamic limit n→∞, viz.,
(44)Pi=limn→∞Pi(n)=1ui1∑j=1N1uj1=11+∑j≠iN1uj1.

The *grand potential*
Ω playing the role of free energy with respect to the grand partition function Z in grand-canonical ensemble equals:(45)$$\Omega =-kT\ln Z=\log_{2}\left(\frac{1}{\gamma_{1}}\sum_{j=1}^{N}\frac{1}{u_{j1}}\right).$$

Having a form of the *relative fugacity* of a node, the grand canonical probability (Equation 44) can be regarded as measuring the *ease of separation of the vertex from the rest of graph* with respect to the entire system of infinite paths. The nodes with the long paths growth rate inferior to the topological entropy are insufficiently integrated into the graph structure and might be lost or acquire new connections in the course of prospective graph structural modifications.

In Figure 6, we have presented the membrane graph (left) and the spatial graph of the city of Lubbock, Texas (right), with their nodes colored according to values of grand canonical probabilities (Equation 44). The nodes located on the low-dimensional graph boundaries, at the corners of membrane graph and in the loosely connected south suburbs of the city of Lubbock have distinctly higher relative fugacity than others. These nodes can also be regarded as the *points of prospective network growth* in where the graph as a system of infinite paths remains open. Interestingly, the highlighted nodes in the spatial graph of Lubbock (Figure 6 right) mark the city neighborhoods currently under construction.

## 9. Discussion and Conclusions

We have defined three major thermodynamic ensembles of ergodic walks in connected undirected graphs, in the thermodynamic limit of infinitely long walks and showed that the ergodic mindset might be applied not only to particles of ideal gases, but also to quite abstract objects of discrete mathematics, such as graphs.

We have demonstrated that graph structural defects and irregularities, such as missing nodes and edges, might dramatically reduce the number of available very long paths, globally reshaping the mobility patterns in the entire graph. In the framework of micro-canonical ensembles, we may consider their effect as resulting from actions of the entropic pressure and force repelling walkers from structural irregularities and boundaries toward the best integrated region of the network: *the laxer the connection, the stronger the repelling*. Perhaps, the cumulative effect of entropic forces generated by railways and other structural obstacles along with the unbalanced growth of urban neighborhoods might be responsible for the urban decay process in the historical districts of some cities.

We have also shown that the problem of effective navigation [36] in graphs and networks can be considered with respect to a canonical ensemble of walks, as an effective location prediction of a navigator’s position, and requires a density of locations in the walk be known. According to the probabilistic setting, frequently visited sites are predicted more efficiently than little frequented, especially in the long run: *the more frequent a node, the more predictable the navigator’s position visiting it*. Regular lattices and homogeneous graphs lacking structural salience and landmarks might also be confusing environments dramatically, reducing predictability of navigator’s position.

Finally, we have studied the grand canonical ensemble of very long paths describing the statistics of fluctuations of the local path growth rate with respect to the graph topological entropy. In the thermodynamic limit of infinite paths, the distribution of the relative fugacity over the graph nodes takes the form of Fermi–Dirac distribution function. The high relative fugacity value of a node assumes that the degree of its integration into the system of infinite paths is insufficient, indicating that *the graph is open for the prospective structural modifications associated with the node*. In the urban spatial graphs, the nodes of high fugacity might be concentrated in the neighborhood under construction, marking the points of city network growth.

Future research should consider a comprehensive structural "equation of sate" for networks and graphs.

## Figures and Tables

**Figure 1 entropy-23-00205-f001:**
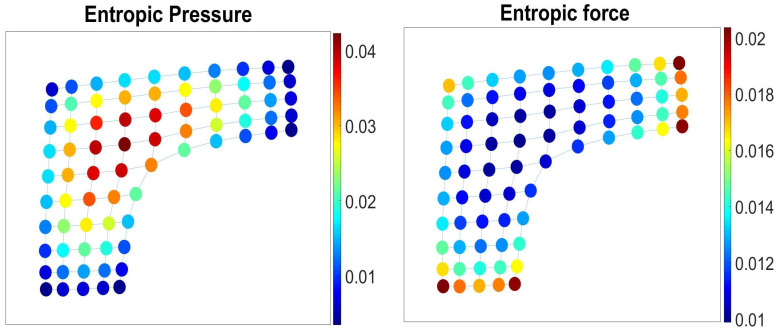
Entropic pressure and force in the membrane graph. **Left**: The nodes are colored according to the values of entropic pressure (Equation 9). **Right**: The nodes are colored according to the values of the Perron eigenvector of the entropic force matrix $F_{ij}$ (Equation 10).

**Figure 2 entropy-23-00205-f002:**
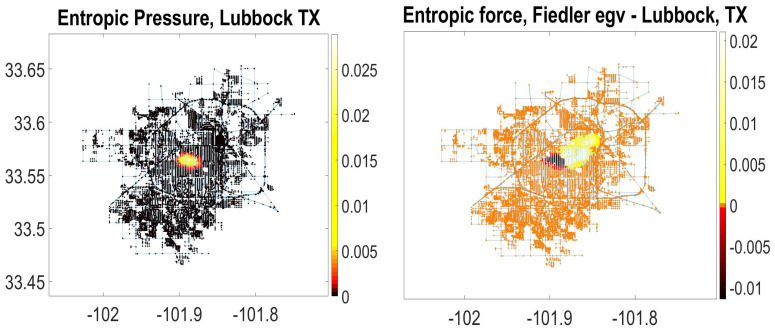
Entropic pressure and force in the city spatial graph of Lubbock, Texas (of 10,421 nodes). **Left**: The nodes of the city graph are colored according to values of entropic pressure (Equation 9). **Right**: The nodes of the city graph are colored according to values of the Fiedler eigenvector belonging to the *second* largest eigenvalue of the entropic force matrix $F_{ij}$ (Equation 10) (or the smallest eigenvalue of the associated Laplacian matrix). The Fiedler eigenvector indicates the direction of fastest decrease of the entropic force over the city spatial graph of Lubbock.

**Figure 3 entropy-23-00205-f003:**
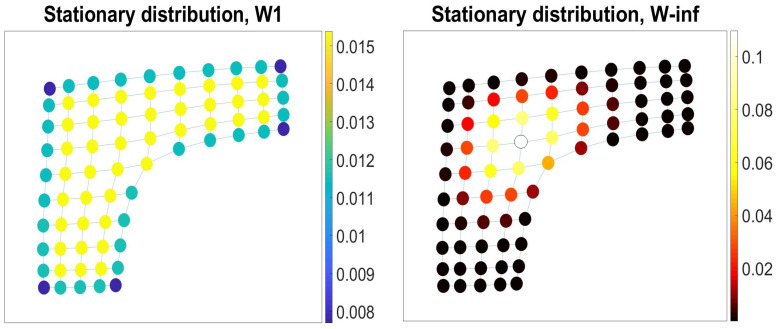
Densities of nodes in the membrane graph with respect to the isotropic and anisotropic intrinsic random walks. **Left**: Density of nodes wrt to the isotropic random walk $W_{ij}^{\left(1\right)}$ is proportionate to their degree centrality. **Right**: The anisotropic random walk $W_{ij}^{\left(\infty \right)}$ is confined in the central nodes of the membrane graph.

**Figure 4 entropy-23-00205-f004:**
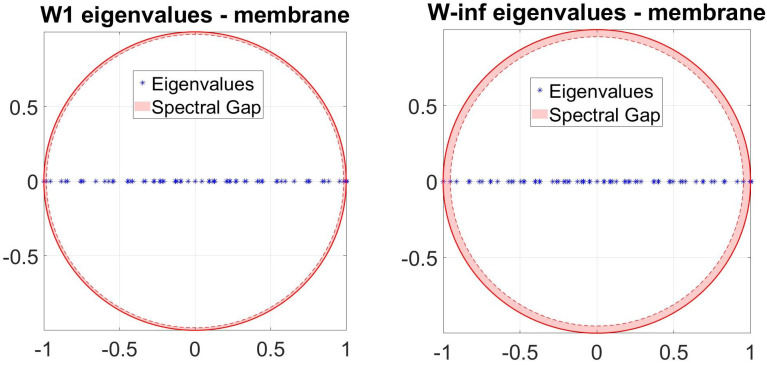
Spectral gaps is maximum (mixing time is minimum) over the canonical ensemble of intrinsic random walks for the anisotropic random walk $W_{ij}^{\left(\infty \right)}$.

**Figure 5 entropy-23-00205-f005:**
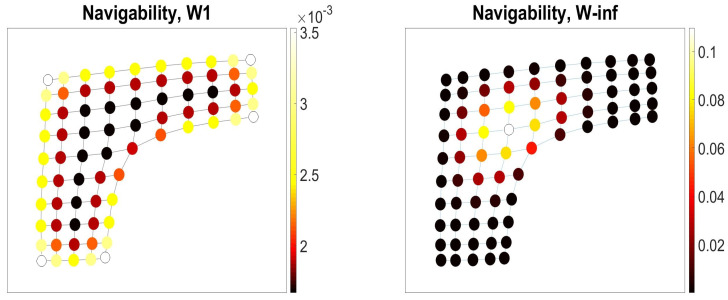
Navigability to the nodes in the membrane graph by the isotropic $W_{ij}^{\left(1\right)}$ (left) and anisotropic $W_{ij}^{\left(\infty \right)}$ (right) intrinsic random walks.

**Figure 6 entropy-23-00205-f006:**
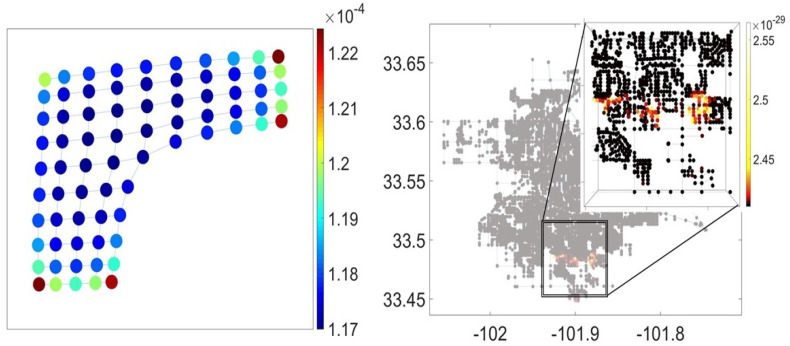
The grand canonical probabilities in the membrane graph (**left**) and in the spatial graph of the city of Lubbock, Texas (**right**). The highlighted nodes exhibit the long paths growth rates inferior to the topological entropy of the graph, in the thermodynamic limit n→∞, and therefore have higher relative fugacity in the course of prospective graph structural changes.

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
