# Peer review of "Infinite Ergodic Walks in Finite Connected Undirected Graphs"

_entropy, 2021, doi:10.3390/e23020205_

Round 1

Author Response

Response to Reviewer 1

We profoundly thank all our referees for their valuable comments, kind suggestions, and thoughtful  illuminating questions which helped us to considerably improve the quality of the manuscript! We have rewritten the abstract, added a leading paragraph, clarifying the subject and motivation of our work, edited and expanded Introduction, rewritten the Discussion and Conclusion section, as well as other sections of our manuscript. Multiple minor corrections have been done to the text.  All figures in the paper have been redrawn to improve the clarity and intelligibility of the presentation.

The detailed response to the review is given below:

“The entropy theory for finite states Markov chain (or random walks on finite graphs) is a mature mathematical theory. See, for example, the books [Denker, Grillenberger, and Sigmund: Ergodic theory on compact spaces. Lecture Notes in Mathematics, Vol. 527. Springer-Verlag, Berlin-New York, 1976. iv+360 pp.] and [Kitchens: Symbolic dynamics. One-sided, two-sided and countable state Markov shifts. Universitext. Springer-Verlag, Berlin, 1998. x+252 pp.]”

Thank you very much for this remark which resulted in the following paragraph added to the Introduction section: “For example, the series of intrinsic

76 random walks (introduced in Sec. 5) making up equal probabilities to all walks of a given

77 length starting at a node provides an example of the canonical ensemble of walks defined on

78 the finite graph. This canonical ensemble contains not only the very well-known isotropic

79 nearest-neighbor random walks on finite graphs [7,10], but also infinitely many types of less

80 known anisotropic random walks on graphs – and the Ruelle-Bowen random walk [12,13] making

81 up all infinite walks starting at each node equally probably is one among them. While the

82 ergodic theory for isotropic random walks on finite graphs is well developed [14,15]1, the

83 ergodic properties of anisotropic random walks, including their statistical confinement in the

84 best structurally integrated sub-graphs (see Secs. 5,7), have not been discussed in literature

85 yet.”

                      1 We profoundly thank our referee for this remark.

(1) Formula (9): It would be better to put n ! 1 under the symbol lim: limn!1 .To do this, one can use \lim\limits instead of \lim in Latex. Please correct many other places.

Thank you very much for your help! The misplacement of limits in the arrays of equations in LaTeX is annoying! All instances have been corrected successfully.

(2) l. 93: Leave a space before \Left".

Done! Thank you very much!

(3) l. 133: Should nr !n!1 n_r be nr n !n!1 _r ?

Corrected! Thank you!

(4) Formula (12): Delete the second equality.

The redundant equality is deleted. Thank you!

(5) Formula (13): For the second equality, after taking the limit, we could

not keep the variable n. To make sense, I suggest not write the limit but

replace all equalities by . There is the same problem for the formula (15).

Thank you for your help! All symbols are corrected accordingly your suggestions.

(6) Formula (33): I do not think the second line is necessary.

The second line is removed – Thank you for your help!

(7) Formula (38): Please check if the formula is correct. Why the first equality holds?

The following exposition has been added after line 225 (please, see the revised pdf text for the intelligible presentation):
As demonstrated in [33], the entropy function I(p) _ H(Xt) allows for the following

decomposition involving the conditional entropies:

H (Xt) _ H (Xt) ? H (Xt+1j Xt) + H (Xt+1j Xt)

= (H(Xt) ? H(Xt+1jXt)) + H (Xt+1jXt) + fH (XtjXt?1) ? H (XtjXt?1)g

+fH (Xt+1jXt?1) ? H (Xt+1jXt?1)g

= (H(Xt) ? H (Xt+1jXt)) | {z }

E(p)

+(H (Xt+1j Xt?1) ? H (Xtj Xt?1)) | {z }

I( Xt;Xt+1jXt?1)

+(H (Xt+1j Xt) + H (Xtj Xt?1) ? H (Xt+1j Xt?1) ) | {z }

U(p)

.

(38)

Therefore, the remaining part of the information function (28), the last part in the

decomposition (38), is the conditional entropy of the present navigator’s location conditioned

on her past and future locations, viz.,

U(p) = I(p) ? P(p) = H (XtjXt?1) + H (XtjXt?1) ? H (Xt+1jXt?1) (39)

assesses the amount of true uncertainty about navigator’s location that can neither be inferred

from integrating over the past history of the p-walk, nor have any repercussion for the

navigator walk in the future [35].”

We profoundly thank our referee again for his invaluable help and important suggestions!

Sincerely,

Dimitri Volchenkov

Reviewer 2 Report

In this submission, the authors have reviewed the microcanonical, canonical, and grand canonical ensembles of very long walks defined in finite connected undirected graphs. Further, they demonstrated that the concept of ergodic ensembles can be applied to abstract objects of discrete mathematics. The submission is scientifically sound and well-written, and therefore I recommend the publication of it in the journal Entropy.

Minor comments:

  • The node values in Figs 5 and 6 are too small.
  • Please fix the width of Fig 6.
  • What is the meaning of colored nodes in Fig 6 (right)?

Author Response

Response to Reviewer 2

We profoundly thank all our referees for their valuable comments, kind suggestions, and thoughtful  illuminating questions which helped us to considerably improve the quality of the manuscript! We have rewritten the abstract, added a leading paragraph, clarifying the subject and motivation of our work, edited and expanded Introduction, rewritten the Discussion and Conclusion section, as well as other sections of our manuscript. Multiple minor corrections have been done to the text.  All figures in the paper have been redrawn to improve the clarity and intelligibility of the presentation.

The detailed response to the review is given below:

  • The node values in Figs 5 and 6 are too small.

Thank you very much! – all related figures in the paper have been redrawn to make them as intelligible as possible. Redundant indices have been removed; the sizes of nodes were increased; bleached nodes have been indicated by the hollow circles. The overall composition  of Figure 6 has been changed.

  • Please fix the width of Fig 6.

The figure has been redrawn and adjusted accordingly to improve visibility and intelligibility of the presentation. We profoundly thank our referee for pointing us at the low quality of figures in the previous version of our manuscript.

  • What is the meaning of colored nodes in Fig 6 (right)?

·         The following explanatory paragraph has been added to the text/edited above the figure: “Having 230 a form of the relative fugacity of a node, the grand canonical probability (44) can be

231 regarded as measuring the ease of separation of the vertex from the rest of graph with respect to

232 the entire system of infinite paths. The nodes with the long paths growth rate inferior to

233 the topological entropy are insufficiently integrated into the graph structure and might be

234 lost or acquire new connections in the course of prospective graph structural modifications.

235 In Fig. 6, we have presented the membrane graph (left) and the spatial graph of the city of

236 Lubbock, Texas (right), with their nodes colored accordingly the values of grand canonical

237 probabilities (44). The nodes located on the low-dimensional graph boundaries, at the corners

238 of membrane graph and in the loosely connected south suburbs of the city of Lubbock have

239 distinctly higher relative fugacity then others. These nodes can also be regarded as the points

240 of prospective network growth in where the graph as a system of infinite paths remains open.

241 Interestingly, the highlighted nodes in the spatial graph of Lubbock (Fig. 6. right) mark the

242 city neighborhoods currently under construction.”

·         The following sentence has been added to the figure caption: “The highlighted nodes exhibit the long paths growth rates inferior to the topological entropy of the graph, in the thermodynamic limit

n ! ¥, and therefore have higher relative fugacity in the course of prospective graph

structural changes.”

We profoundly thank our referee again for his invaluable help and important suggestions!

Sincerely,

Dimitri Volchenkov

Reviewer 3 Report

The paper introduces standard thermodynamics terminology into a slightly novel context. However, it does so clumpsily, and only to solve tasks that can already be solved with standard (probability theoretic) machinery as well. The research idea is not without merit, but the quality of the results do not motivate the heavy machinery introduced.

Author Response

Response to Reviewer 3

We profoundly thank all our referees for their valuable comments, kind suggestions, and thoughtful  illuminating questions which helped us to considerably improve the quality of the manuscript! We have rewritten the abstract, added a leading paragraph, clarifying the subject and motivation of our work, edited and expanded Introduction, rewritten the Discussion and Conclusion section, as well as other sections of our manuscript. Multiple minor corrections have been done to the text.  All figures in the paper have been redrawn to improve the clarity and intelligibility of the presentation.

The detailed response to the review is given below:

The paper introduces standard thermodynamics terminology into a slightly novel context. However, it does so clumpsily, and only to solve tasks that can already be solved with standard (probability theoretic) machinery as well. The research idea is not without merit, but the quality of the results do not motivate the heavy machinery introduced.

Thank you very much for the criticism that inspired our efforts to highlight our results and further explain the motivation of our work in more details!

1. The abstract has been rewritten in the following way:

The micro-canonical, canonical, and grand canonical ensembles of walks defined in finite connected undirected graphs are considered in the thermodynamic limit of infinite walk length. As infinitely long paths are extremely sensitive to structural irregularities and defects, their properties are used to describe the degree of structural imbalance, anisotropy, and navigability in finite graphs. For the first time, we introduce entropic force and pressure describing the effect of graph defects on mobility patterns associated with the very long walks in finite graphs; navigation in graphs and navigability to the nodes by the different types of ergodic walks; as well as node’s fugacity in the course of prospective network expansion or shrinking.”

2. The following leading paragraph has been added to the opening of the manuscript:

The precursor of a concept of statistical ensembles and the related ergodic hypothesis formulated by Boltzmann [1,2] were met violently negative by the great majority of scientists for clumsiness, absurd, and paradoxical consequences [3] although it allowed the theoretical calculation of the equations of state for the first time. The study of statistical ensembles related to graphs and networks suffers the similar inhospitable reception from scientists playing cup-and-ball with a swarm of heuristic parameters and giving any importance to their connection with each other that is often responsible for spurious conclusions on graph’s structure and function. The thermodynamic approach to graphs was initiated in complex network theory concerned with the thermodynamic limit of infinitely large graph size N ! ¥ [4], in which graph’s structural "fluctuations" become negligible. The major result of the theory on structurally homogeneous infinite graph (random trees) is the Bose–Einstein condensation mechanism explaining the growth of complex evolving networks as a topological phase transition between a "rich-get-richer" phase and a "winner-takes-all" phase [57]. In contrast to complex network theory, we consider the statistical ensembles of walks defined on a finite connected undirected graph in the thermodynamic limit of very long walks n ! ¥, which has previously never been addressed. Statistics of lengthy walks elucidates the graph structure, quantify navigability of the graph, and evaluate the fugacity of graph nodes with respect to the entire system of infinite paths available in the graph - all these characteristics are introduced and discussed in our work for the first time. The probability measuring the tendency of a graph to shrivel or expand at a node follows the Fermi-Dirac distribution function. Although we have sketched a set of "ideal gas laws" for the structure of networks and graphs (in the last section of our work), we have not formulated a comprehensive structural "equation of sate" for graphs and networks yet.”

3.) The following paragraph has been added to the introduction section:

For example, the series of intrinsic random walks (introduced in Sec. 5) making up equal probabilities to all walks of a given length starting at a node provides an example of the canonical ensemble of walks defined on the finite graph. This canonical ensemble contains not only the very well-known isotropic nearest-neighbor random walks on finite graphs [7,10], but also infinitely many types of less known anisotropic random walks on graphs – and the Ruelle-Bowen random walk [12,13] making up all infinite walks starting at each node equally probably is one among them. While the ergodic theory for isotropic random walks on finite graphs is well developed [14,15]1, the ergodic properties of anisotropic random walks, including their statistical confinement in the best structurally integrated sub-graphs (see Secs. 5,7), have not been discussed in literature yet.”

4.) The following paragraph has been introduced in Sec 7 about the navigability:

Navigability to a node evaluated by the partition function (37) depends on the strategy of walkers. In Fig. 5, we illustrate the difference by highlighting the nodes of membrane graph accordingly their degrees of navigability by the isotropic random walks W(1) ij (left)  and anisotropic random walks W(¥) ij (right). For the isotropic random walks, the movement of walkers along the low-dimensional boundaries and at the corners of the graph are more predictable than their movements in the bulk, as all bulky locations of the same connectivity are visited with equal probability by the random walk W(1) ij . In contrast with the isotropic random walks, a navigator following the anisotropic strategy W(¥) ij is statistically confined within the region hosting the most of infinitely long paths available in the graph, where navigator’s position is very likely.”

5.) The following paragraph has been introduced in Sec 8 about the relative fugacity:

Having a form of the relative fugacity of a node, the grand canonical probability (44) can be regarded as measuring the ease of separation of the vertex from the rest of graph with respect to the entire system of infinite paths. The nodes with the long paths growth rate inferior to the topological entropy are insufficiently integrated into the graph structure and might be lost or acquire new connections in the course of prospective graph structural modifications. In Fig. 6, we have presented the membrane graph (left) and the spatial graph of the city of Lubbock, Texas (right), with their nodes colored accordingly the values of grand canonical probabilities (44). The nodes located on the low-dimensional graph boundaries, at the corners of membrane graph and in the loosely connected south suburbs of the city of Lubbock have distinctly higher relative fugacity then others. These nodes can also be regarded as the points of prospective network growth in where the graph as a system of infinite paths remains open. Interestingly, the highlighted nodes in the spatial graph of Lubbock (Fig. 6. right) mark the city neighborhoods currently under construction.

6.) The Discussion and Conclusion section has been rewritten in the following way:

We have demonstrated that graph structural defects and irregularities, such as missing nodes and edges, might dramatically reduce the number of available very long paths, globally reshaping the mobility patterns in the entire graph. In the framework of micro-canonical ensembles, we may consider their effect as resulting from actions of the entropic pressure and force repelling walkers from structural irregularities and boundaries toward the best integrated region of the network: the laxer connection, the stronger repelling. Perhaps, the cumulative effect of entropic forces generated by railways and other structural obstacles along with the unbalanced growth of urban neighborhoods might be responsible for the urban decay process in the historical districts of some cities. We have also shown that the problem of effective navigation [19] in graphs and networks can be considered with respect to a canonical ensemble of walks, as an effective location prediction of navigator’s position requires a density of locations in the walk be known. Accordingly the probabilistic setting, frequently visited sites are predicted more efficiently than little frequented, especially in the long run: the more frequent a node, the more predictable navigator’s position visiting it. Regular lattices and homogeneous graphs lacking structural salience and landmarks might also be confusing environments dramatically reducing predictability of navigator’s position. Finally, we have studied the grand canonical ensemble of very long paths describing the statistics of fluctuations of the local path growth rate with respect to the graph topological entropy. In the thermodynamic limit of infinite paths, the distribution of the relative fugacity over the graph nodes takes the form of Fermi-Dirac distribution function. The high relative fugacity value of a node assumes that the degree of its integration into the system of infinite paths is insufficient, indicating that the graph is open for the prospective structural modifications associated to the node. In the urban spatial graphs, the nodes of high fugacity might be  concentrated in the neighborhood under construction, marking the points of city network growth.Future research should consider a comprehensive structural "equation of sate" for networks and graphs.”

We profoundly thank our referee again for his invaluable help and important opinion!

Sincerely,

Dimitri Volchenkov

Round 2

Reviewer 3 Report

I am still not convinced about the significance of this approach to graph connectivity. However, the text has been edited as to much better motivate the results, and I believe the paper is now worthy of publication.